# Bioactive Glass-Enhanced Resins: A New Denture Base Material

**DOI:** 10.3390/ma16124363

**Published:** 2023-06-13

**Authors:** Zbigniew Raszewski, Katarzyna Chojnacka, Marcin Mikulewicz, Abdulaziz Alhotan

**Affiliations:** 1SpofaDental, Markova 238, 506 01 Jicin, Czech Republic; 2Department of Advanced Material Technologies, Wroclaw University of Science and Technology, 50-370 Wroclaw, Poland; katarzyna.chojnacka@pwr.edu.pl; 3Department of Dentofacial Orthopedics and Orthodontics, Division of Facial Abnormalities, Medical University of Wroclaw, 50-367 Wroclaw, Poland; marcin.mikulewicz@umw.edu.wroc.pl; 4Department of Dental Health, College of Applied Medical Sciences, King Saud University, Riyadh P.O. Box 12372, Saudi Arabia; aalhotan@ksu.edu.sa

**Keywords:** acrylic resin, mechanical properties, bioactive glass, ions releasing, hydroxyapatite

## Abstract

Background: The creation of the denture base material with bioactive properties that releases ions and produces hydroxyapatite. Methods: Acrylic resins were modified by the addition of 20% of four types of bioactive glasses by mixing with powders. Samples were subjected to flexural strength (1, 60 days), sorption and solubility (7 days), and ion release at pH 4 and pH 7 for 42 days. Hydroxyapatite layer formation was measured using infrared. Results: Biomin F glass-containing samples release fluoride ions for a period of 42 days (pH = 4; Ca = 0.62 ± 0.09; P = 30.47 ± 4.35; Si = 22.9 ± 3.44; F = 3.1 ± 0.47 [mg/L]). The Biomin C (contained in the acrylic resin releases (pH = 4; Ca = 41.23 ± 6.19; P = 26.43 ± 3.96; Si = 33.63 ± 5.04 [mg/L]) ions for the same period of time. All samples have a flexural strength greater than 65 MPa after 60 days. Conclusion: The addition of partially silanized bioactive glasses allows for obtaining a material that releases ions over a longer period of time. Clinical significance: This type of material could be used as a denture base material, helping to preserve oral health by preventing the demineralization of the residual dentition through the release of appropriate ions that serve as substrates for hydroxyapatite formation.

## 1. Introduction

The first stage of colonization by microorganisms is the adsorption of salivary membrane proteins by all accessible surfaces of the oral cavity and the acrylic surface of the prosthesis. On this surface, adhesion and growth of microorganisms occur. When microorganisms are appropriately accumulated, they form structures known as biofilms, which are highly organized microbial communities entangled in a three-dimensional matrix. This structure confers many benefits to colonizing species, such as antimicrobial and host defense, increased coaggregation and interaction properties [1]. However, biofilms formed on dentures are different in terms of bacterial colonization have higher content of *Streptococcus mutans*, *Streptococcus mitis*, and *Streptococcus oralis* compared to dental plaque. This can be observed as early as 24 h after placing the restorations in the oral cavity, and a mature biofilm forms after 72 h [2,3]. *S*. *aureus* and *C*. *albicans*, which are often found on the surface of dentures, can cause the transformation of homeostatic biofilm into a dysbiosis biofilm, which is already directly responsible for various types of diseases [4]. The literature states that more than 70% of difficult-to-treat and persistent infections are caused by microorganisms growing in biofilms [5].

Biofilm often accumulates on the surface of acrylic dentures during use (due to the absence of ionic charge), which leads to the formation of calculus over time. This problem may occur both in prostheses made using the traditional method, CAD CAM technology, or 3D printing [6]. It becomes an area where pathogenic bacteria multiply, which can cause various types of diseases within the body including demineralization of teeth, dental caries, gingivitis, periodontitis, periapical periodontitis, and peri-implantitis [7]. The process of colonization of acrylic restorations that are fixed on implants may lead to changes in the surface of the implant and the host tissues [8]. That is why it is desirable to create a material that will prevent the formation of biofilm on the surface of acrylic dentures. This is possible in three directions: antimicrobial agent release, contact-dependent strategies, and multifunctional strategies.

Application of an antimicrobial agent polyhexamethylene guanidine hydrochloride (PHMGH) inhibits biofilm formation on resinous materials against *S*. *mutans* [9]. Another way to obtain antibacterial properties is to use cationic resins at a concentration of 2.5–10% mass fraction of dimethylamino dodecyl methacrylate (DMADDM) which improves the antibacterial effect expressed in *S. mutans* and limits demineralization of the tested resin [9]. Greater effectiveness can be achieved when DMAHDM reacts with 2-methacryloyloxyethyl phosphorylcholine [10]. The same compound has also been shown to be effective as a reaction product with glass ionomer cement.

Another approach, known as contact-dependent strategies, involves creating a material with bactericidal properties by incorporating nanoparticles such as zinc oxide [11], zirconium dioxide, and silver nanoparticles [12,13], or silver vanadate (AgVO3) [14]. Nanomaterials, graphene oxide nanosheets, and carbon nanotubes were also successfully used, in which the mechanism of action is direct contact bacteria killing properties. This effect was achieved by applying a 2% supplement for 28 days [15]. The expansion of the concept of metals and the surface of acrylic plastic can also be modified by the addition of various types of metal salts with acrylic acid, zirconium methacrylate, tin methacrylate, and di-n-butyl methacrylate-tin [16]. The texture of the surface, as a result of material polishing, can reduce the adhesion of the microorganism and the formation of biofilm. This is due to changes in the surface of the denture and leads to an increase in the silicon atom concentration and a decrease in surface carbon [17]. Further modification of the surface can be achieved by interaction of plasma. Plasma-modified PMMA samples have 1.5–2.5 times lower microbial adhesion compared to unmodified samples, while their surface free energy increases up to 1.5 times due to the formation of additional polar oxygen-containing chemical groups induced by the plasma. The surface of the prosthesis subjected to plasma treatment showed good biocompatibility and less irritating effect compared to unmodified surfaces [18].

Another approach to modify the surface is use of different types of coatings (Shibata). studied the use of poly (2-methacryloyloxyethyl phosphorylcholine-co-n-butyl methacrylate), which drastically reduced the ability of cariogenic bacteria, such as *S*. *mutans* and *S*. *sobrinus*, to develop biofilms [19]. Finally, a further approach is to modify the surface to have a negative charge, which can be achieved with polyacrylic acid and poly itaconic acid, which have been tested as surface treatments on conventional denture base materials. Both acids demonstrated significant inhibition of *C*. *albicans* growth. The use of carboxyl groups by applying their coatings reduced the adhesion of *C*. *albicans* by 90%, which was tested by Acosta et al. [20]. However, using coatings has certain limitations, as the active layer can be removed under the influence of food or hygiene procedures (denture disinfectants, toothpaste, and toothbrush). Therefore, modifications of the entire material seem to be more promising. In the case of our team, bioactive glasses were added to traditional thermally polymerized PMMA resin [21,22]. We managed to obtain a material that was capable of releasing calcium, phosphorus, silicon, and ions for a period of 42 days. However, the amount of these ions was significantly reduced with time. The second problem was the reduction of mechanical properties (glasses were not chemically bonded with PMMA) [22]. To solve this problem, it was decided to use two variants of the same glass like in the previous study. One part has been silanized and the other has not been modified (50/50). The mixture of glasses improved in this way was added at a concentration of 20% to the heat curing denture base material. The thesis put forward at the beginning of this study is that by using modified glasses, the material will be able to release ions over a longer period of time in a more even manner. In addition, the use of the silanization process will improve the mechanical properties of the obtained material.

The paper’s findings hold significant clinical implications, as the development of such denture base materials could improve patient outcomes by maintaining oral health and preserving residual dentition. The incorporation of partially silanized bioactive glasses may pave the way for further innovations in dental prosthetics and set the stage for future research aimed at optimizing and customizing these materials for individual patient needs.

## 2. Materials and Methods

Samples of acrylic resin polymerized hot curing method by the addition of bioactive glasses (totally 104 samples) were used for the tests. The same resin for making denture base (totally 30th sample) polymerized according to the manufacturer’s recommendations was used as a reference sample. A detailed description of the sample’s preparation is given below, and all performed tests are summarized in Figure 1.

### 2.1. Silanization of Bioactive Glasses

Glasses used in our previous work (Biomin F, Biomin C, 53P4, and 45S5 (all samples were delivered by Cera Dynamic England, raw composition is presented in Table 1. For the silanization process, powder in the amount of 20 g was mixed with a solution of 90% ethyl alcohol containing 1% gamma trimethoxysil propyl methacrylate and 0.5% concentrated acetic acid (all raw materials from Sigma Aldrich, Praha, Czech Republic). The entire suspension was stirred with a magnetic stirrer for 60 min (Fisher Scientific, Praha, Czech Republic, 400 rpm). Then, it was washed with alcohol and water 3 times. The samples were dried at 105 °C for 24 h to complete the silanization process. The acrylic powders are then ground in a laboratory mortar and sieved through a 100-mesh sieve (Merck, Gernsheim, Germany).

### 2.2. Preparation of a Sample of Resin Modified with Bioactive Glass

The acrylic powder Superacryl Plus (SpofaDental, Jicin, Czech Republic, batch number 567,823) in an amount of 80 g was mixed with 10 g of the appropriate nonsilanized glass and 10 g of silanized glass. To obtain a homogeneous mixture, the whole was placed in a porcelain ball mill (Izerska Porcelanka, Praha, Czech Republic), rotational speed 40 rpm. Number of balls 200 g, diameter 5 mm. The powders were then mixed with the Superacryl Plus liquid at a ratio of 2.4 g of powder to 1 g of monomer. The samples were placed in vessels for 10 min—dough time. When the dough was not sticky to the hand, it was placed in metal molds. For the determination of flexural strength, the samples had a length of 50 × 3 × 10 mm. For the testing of sorption, solubility and ion release, the material was placed in a mold with a diameter of 15 mm and a thickness of 2 mm.

The molds were placed in a manual laboratory press, under a load of 2000 kg, for a period of 10 min to press out excess dough. Resin samples were thermally polymerized in water, initially for 30 min at 60 °C and then for 60 min at 100 °C. After the curing process was completed, the polymerization frame was opened, and the samples were removed. Then, their edges were smoothed with sandpaper (120, Saint Gobain, Kolo, Poland).

### 2.3. Flexural Strength

Samples to test this parameter were placed in distilled water in sealed PE containers in a dryer at a temperature of 37 °C. The first six samples were subjected to a three-point fracture test after 24 h using a Shimadzu compressive strength instrument (AGS 10 kNG, Shimadzu, Kyoto, Japan), the width of the supports was 50 mm and the speed of the breaking head was 5 mm/min. The test ended with the fracture of the sample. For the first series of tests, 6 samples were used, and the next 6 samples were stored for 60 days in the same conditions. Distilled water was changed every 4 days. A second batch of samples was subjected to the same test after a period of 60 days. In total, 60 samples were made for the entire study. Pure polymethyl methacrylate (PMMA Superacryl Plus, SpofaDental, Jicin, Czech Republic batch 567,823) polymerized under the same conditions was used as reference material, and according to user instruction [20]. A detailed description of the tests and the amount of sample needed for testing are provided in ISO 20795-1: 2013 (EN) [23].

### 2.4. Sorption and Solubility

Sorption and solubility were analyzed according to ISO 20795-1: 2013 (EN), Dentistry—Denture base polymers [23]. Samples of the polymerized material (15X 2 mm, six of each composition, 30 totally) were placed in a desiccator and weighed every other day until a constant weight was obtained *(M1*). Then, they were placed in distilled water for 7 days. Once removed from the water, the discs were dried with a paper towel and reweighed on an analytical balance with an accuracy of 0.0001 g (Precioza 256, Turin, Italy), (*M2*). The samples were once again placed in a desiccator filled with molecular sieves (Sigma Aldrich, Poznan, Poland) and weighed until a constant mass of *M3* was reached. Pure acrylic resin was used as a reference sample. The solubility and solubility of the materials were determined according to the following equations.

*A*—sorption, *B*—solubility.

These two parameters were calculated from the following Equations (1) and (2):(1)$$A=\frac{M2-M1}{S}$$
(2)$$B=\frac{M1-M3}{S}$$
where *M2* is the mass of the sample after 7 days in distillate water, *M1* is the mass before immersion in water, and *M3* is the mass drying of the material in the exicator after immersion in water. The S is the surface of the disc, measured by calibrated caliper [21].

### 2.5. Assessment of Ion Release of Glass and Acrylic Resins in Artificial Saliva

To assess the bioactive properties of the samples, we tested ions released into artificial saliva at pH 4 and 7 for 1, 28, and 42 days. The artificial saliva solution was prepared by dissolving sodium chloride (0.4 g) (NaCl, Sigma Aldrich, Poland), potassium chloride (1.21 g) (KCl, Sigma Aldrich, Poland), hydrated potassium dihydrogen phosphate (0.78 g) (NaH_2_PO_4*_ 2H_2_O, Sigma Aldrich, Poland), hydrated sodium sulfide (0.12 g) (Na_2_S_*_ 9H_2_O, Sigma Aldrich, Poland), and urea (1.0 g) (Sigma Aldrich, Poznan, Poland) in ultrapure water (1000.0 g) (Merck, Gernsheim, Germany). The prepared solution was transferred to two vessels and adjusted to pH 4 and 7 using hydrochloric acid (0.1 Mol) and sodium hydroxide (0.1 Mol), respectively (both from Merck, Gernsheim, Germany), as in previous of our study [22]. Fifteen samples were tested, including three of each type of glass and acrylic resin, as reference materials. The disks, 15 mm in diameter and 2 mm thick, were prepared according to a previously described method.

Sample extracts from artificial saliva were analyzed by inductively coupled plasma atomic emission spectrometry (ICP-AES) using an iCAP 6500 Duo optical spectrometer (Thermo Fisher Scientific, Waltham, MA, USA). The extraction process was completed after 1, 28, and 42 days, respectively. After the extraction process, the samples were acidified with trace pure nitric acid and made up to 20 mL. The blank and extraction samples were then used for multielement analysis.

To determine the fluoride ion concentration, the extract from the dental material was directly injected through a sterile 0.2-μm syringe filter before entering the chromatography column. The concentration of fluoride ions was measured using a Dionex ICS 1100 ion chromatograph (Thermo Fisher Scientific, Waltham, MA, USA). One-way ANOVA and the Tukey HSD test calculator on Astasta.com were used for statistical analysis, with a confidence level of *p* < 0.05.

### 2.6. Hydroxyapatite (HA) Formation

Pure acrylic resin samples and samples containing bioactive glasses, which were used for solubility and solubility tests, were subsequently used for surface testing using an IR spectrophotometer. The samples were placed on the window of this instrument, Nicolet I5S (Thermoscientific, Prague, Czech Republic) and measured in the IR range of 4000–500 cm^−1^. For each of the 5 sample groups, 3 measurements were made.

The scheme of the tests performed in this research is presented in Figure 1 in the form of a graphic abstract.

### 2.7. Vickers Hardness Measurement

To test the hardness of the material, 20 samples were used, which were obtained from the bending resistance test after 60 days of storage in distilled water. This parameter was assessed with a digital microhardness tester (FM-700, Future Tech Corp., Kawasaki, Japan). An indenter point in the form of a square-based pyramid was applied at a load of 300 g for 15 s at room temperature at 37 °C. Five indentations were made at different points along each specimen on the same surface side, with a minimum distance of 1 mm between any two indentations. The mean hardness value of each specimen group was then calculated. The Vickers microhardness (HVN) value was calculated using Equation (3)
(3)$$HV=1.854\left(\frac{F}{D2}\right)$$
with *F* being the applied load (measured in kilograms-force) and *D2* the area of the indentation (measured in square millimeters).

### 2.8. Statistical Analysis

The results were statistically analyzed using a one-way analysis of variance at a significance level of 0.05, as a reference using a sample of PMMA resin not modified with fillers. In addition, a post hoc analysis was performed by using Tukey’s HSD test (using a free test calculator provided by Astatsa, San Jose, CA, USA).

## 3. Results

### 3.1. Flexural Strength

The results of flexural strength samples stored in distilled water for 24 h and 60 days are shown in Table 2.

It should be noted, however, that all samples after storage in distilled water, both after 24 h and 60 days, have a flexural strength greater than 65 MPa. This means they meet the ISO 20795 Denture Base Polymers standard [22]. After storage in distilled water for 2 months, a decrease in flexural strength was observed from 11% (for Biomin C and Biomin F) to 18% (45S5, 53P4).

### 3.2. Sorption and Solublity

Another important material property is solubility and sorption, which indicate whether the material can extract ions from its composition (solubility) and to what extent water will penetrate into it (sorption). The results of this study after 7 days of storage in distilled water are presented in Table 3.

The highest solubility value was found for Biomin C—2.6 *±* 0.88 µm/mm^3^. For others, the solubility values were similar to that of pure PMMA (reference sample). The higher sorption was obtained for samples containing 20% 53P4 glass (19.15 *±* 2.37 µg/mm^3^).

### 3.3. Ions Releasing

Samples containing Biomin F after modification of half of the glass with silanes can release fluoride ions for a longer period of time (60 days). In a previous study, the secretion of this ion proceeded very rapidly during the first 24 h and then decreased to zero [22]. A gradual release was obtained for these two glasses also in the case of the release of calcium, phosphorus, and silicon ions over time, regardless of pH = 4 and pH = 7. Results from these tests are presented in Table 4.

### 3.4. Vickers Hardness

The surface hardness test of the sample is shown in Table 5.

The addition of bioactive glasses in the amount of 20% (50/50 silanized/non) causes a slight 1–2 unit increase in the material’s hardness.

### 3.5. Hydroxyapatite Formation

The formation of this layer of material on the pores after storage in distilled water at 37 °C after 7 days was examined by IR spectra (Figure 2).

High absorption in the range of 3500 cm^−1^ indicates the presence of OH^–^ groups from hydroxyapatite. Furthermore, samples with bioactive glasses show absorption at 1460 cm^−1^ (CO_3_^2−^), 1041 cm^−1^ (corresponding to the PO_4_^3−^ vibration), and 570 cm^−1^ (indicating the PO_4_^2−^ group). Comparing these results with standard spectra from the library, it is evident that a layer of hydroxyapatite is formed on the surface of the samples stored in distilled water for 7 days.

## 4. Discussion

Modification of acrylic materials runs constantly and proceeds basically in two directions. The first is to increase their mechanical properties and the second is to increase their biological or bactericidal properties. Modifications that improve the mechanical properties are primarily additions of ZrO_2_ nanomaterials [24,25], aluminum oxide [26], and cerium oxide [27].

Antibacterial properties can also be obtained by the addition of, e.g., AgVO_3_ [14], mesoporous silica nanoparticles [28], zinc-modified phosphate-based glass microfiller [29], and ZnO [30].

Bioactive properties in the case of glasses can be classified as the possibility of releasing calcium, phosphorus, and fluoride ions, which can form a new layer of hydroxyapatite and have bactericidal properties or reduce adhesion to the surface of the material, e.g., in relation to *Candida albicans* (glass 45S5) [12].

The release of ions from various dental materials is a highly desirable property that can ensure their better biocompatibility. In this context, there are already a number of products on the market for filling crown and root canals, which release strontium ions and silicon, prereactive glass ionomer filler [31].

The thesis put forward at the beginning of this investigation has been partially confirmed. Materials such as Biomin C and S53P4 and 45S5 release silicon and phosphorus ions more uniformly, and fluoride ions through Biomin F. Mechanical properties have slightly deteriorated despite the use of 20% glass in the composition of the PMMA resin.

The release of calcium ions in Biomin F, 53P4, and 45S5 glasses is very fast. Different values are obtained for Biomin C, which releases these ions evenly over a long period of time, regardless of the pH of the solution. Similar results were obtained in our previous work [22]. Therefore, silanization does not affect the uniform release of calcium ions in Biomin F, 45S5, and S53P4 glasses. However, the Biomin C sample releases these ions in a uniform and more homogeneous manner when half of the silanized glass is inside.

The fluoride ions contained in Biomin F glass are released gradually and uniformly throughout the test period at pH = 4 and pH = 7. This is a difference from our previous work in which the same glasses added to acrylic were not silanized. Then, in a solution of pH = 4, all the fluoride was quickly washed out within 1 day. This is similar to the direct addition of fluorine compounds, e.g., NaF, to acrylic resin causes a very fast release of this ion within 1–7 days [32].

The influence of the silanization process (the pH of the solution) on the release of fluoride ions was described in the paper Nakornchai [33], for glass G018-090 and Piyananjaratsri [34].

In the case of Biomin F glass, which has been silanized in an acetic acid environment, after this step, a thin layer of polysalt matrix is formed on the surface of the fluor aluminosilicate glass. The matrix layer, which consisted of calcium and aluminum acetates and fluoride ions, was easily penetrated by water. Therefore, the leaching of fluoride ions from the surface of the so-modified matrix into the aqueous solution may be easier than the surface of the glass filler itself [32]. These authors have concluded that acrylic resin can release fluoride ions for 56 days for glasses [32] or 15 days for dopped ions to acrylic resins [33,34,35].

Low fluoride concentration ranged from 0.024–0.154 ppm/mL, reduced demineralization of the enamel surface [36]. In the case of Biomin F glass, the amount of fluorine ions released was 3 ppm/mL and remained at a constant level for a period of 42 days. Fluoride ions released in such quantities may have a bactericidal effect [36]. Therefore, we can assume for more tests in the future that this material will have properties that prevent demineralization of the enamel of the residual dentition, having direct contact with the denture base made with the addition of this type of glass.

Phosphorus ions are uniformly extracted in all test glasses at a constant level over a period of 42 days. Values of 20–30 mg/L are almost twice as high as those obtained in previous studies [22], which proves that the silanization process prolongs the period of releasing ions to artificial solutions.

The release of ions from the PMMA/glass material can be explained in three steps. At the beginning, the ions contained on the surface of the material pass into the solution. Next is washing out the residual monomer and other components with water from the resin. Some free spaces begin to form in the structure of the material, through which water can penetrate. It creates the possibility for ions migration by gradually hydrolyzing the glasses inside the resin, which explains why the materials are able to release these ions over a longer period of time. Since silanes are hydrophobic molecules, they significantly slow down this process [24,37]. This has been confirmed in these studies.

If the amount of unsilanized glass added is increased, a decrease in flexural strength can be observed [29]. For this reason, the measurable benefit of using half of the glass in the silanized form is the improvement of the mechanical properties of the new sample (Agarwal) [35]. They are lower than pure PMMA (reference sample), but at the same level as in previous studies, when the sample was filled with glass at 10%. At the moment, it is 20% concentration of glass in the acrylic resin [22]. The results of flexural strength of acrylic resins in the case of bioactive glasses tested in our case are in line with [32]. Silanized samples have better resistance to breakage even after 2 months of storage in distilled water. These values in the case of Japanese authors range from 72–76 MPa [32].

The amount of released ions will also be affected by the solubility of the material, i.e., the penetration of water into the resin. In the case of samples containing Biomin C, the largest amount of released calcium, phosphorus, and silicon ions was observed, which is accompanied by the highest solubility of this material among the tested samples.

The silanized glasses used in these studies reduced the solubility of the glasses in relation to the same PMMA/glass system from our previous work. The sorption has not changed. What can be concluded is that the process of silanization is an important process that increases the degree of cross-linking of the sample, which reduces their solubility (F) [38]. The important thing is that the flexural strength for all samples, even over a long period of 60 days, is greater than 65MPa. This means that all materials meet this requirement described in the ISO 20795-1:2013 Dentistry-denture base polymers standard [23].

Acrylic resins are relatively soft materials, which, under the influence of hygiene procedures or consumed food, may be subject to local abrasion. The polished surface at the beginning, after the manufacture of the prosthesis, is roughened, which is the precondition for colonization by microorganisms and the formation of a biofilm. Therefore, it is important that the material has the right hardness. In addition, a soft surface makes it easier to absorb dyes from food, which changes the color of the used restoration. Thus, the Vickers microhardness test is considered to be a valid method to evaluate rigid polymers [39].

The addition of bioactive glasses slightly increases the Vickers hardness from 12.87 ± 0.27 to 15.36 ± 0.60 HV. These are in line with the results obtained by a Farina [39], who obtained values of 15–17 HV for thermally polymerized materials. Higher hardness (18.57 HV) was obtained by Duymus for heat curing resin [40]. The differences may be due to the fact that other authors tested the material after polymerization. In this study, the samples were stored in water for 60 days. Water absorption causes a plasticizing effect and thus a decrease in the hardness of the material.

Immersion of acrylic samples containing bioactive glasses in the water causes that, under the influence of time, a layer of hydroxyapatite forms on their surface, which has been proved in these studies using IR spectroscopy. Hydroxyapatite can also form not only on the surface of the modified denture, but also on the residual dentition, which is in direct contact with the prothesis, which prevents demineralization of the enamel. In addition, the existence of this phenomenon has been confirmed by other authors in the case of composite materials [41,42].

However, the research conducted above has some limitations. In order for the material for the denture base to be created, further research on the analysis of the amount of hydroxyapatite produced, biological tests (cytotoxicity and others) and the study of the strength of the connection between the teeth and the denture plate and the content of residual monomers in the polymerized material are necessary, as the use of glasses can significantly affect all these parameters.

### Future Perspectives

The results of this study demonstrate the potential of using partially silanized bioactive glasses in the development of acrylic resin materials for denture base applications. These materials exhibit promising ion release properties and mechanical strength, which could contribute to the prevention of demineralization and support the formation of hydroxyapatite in residual dentition. To fully comprehend the implications and possibilities of this technique, several issues need for more study and development. The long-term repercussions of the release of ions are a crucial factor to consider. Future research should prolong the observation time to explore the long-term ionization behavior, mechanical properties, hygiene and any possible effects on oral health. This study examined the ionization characteristics of materials over a 42-day period. Future research should examine these materials’ cell toxicity and biocompatibility both in vitro and in vivo in order to confirm their safety for clinical usage. This will assist in determining whether the substance is safe for prolonged interaction with oral tissues and whether any negative responses are possible.

Another crucial aspect is the integration with dental prosthetics. Further research should focus on evaluating the bond strength between the teeth and the denture plate made from these modified acrylic resin materials. The success of dental prosthetics depends on the stability and durability of this connection, and it is essential to ensure that the addition of bioactive glasses does not compromise this aspect.

Future studies should explore the content of residual monomers in the polymerized material. The presence of residual monomers can affect the mechanical properties and biocompatibility of the denture base material, making this a crucial parameter to investigate.

The surface characteristics of these modified acrylic resins should also be investigated, with an emphasis on wettability, roughness, and the impact on the development of biofilms. This will give important information on how these materials could affect oral hygiene and the general health of the oral cavity.

Future investigations should also consider exploring the possibility of tailoring the material properties by adjusting the composition and percentage of bioactive glasses in the acrylic resin. This would allow for the development of customized denture base materials that cater to the specific needs of individual patients.

The use of this type of bioactive glass in further research can be extended to materials used in CAD CAM technology and 3D printing [6].

## 5. Conclusions

The addition of bioactive glass Biomin F to the acrylic resin allows for a continuous release of fluoride ions over a period of 42 days.Samples containing Biomin C release a large amount of ion, phosphate, and silicate anions.The mechanical properties of acrylic resins that contain 20% of bioactive glasses (50/50 silanized or not) meet the flexural strength normative requirements for denture plate materials.On the surface of the sample, using the IR technique, it was possible to identify the formation of hydroxyapatite under the influence of storing the sample in distilled water.

## Figures and Tables

**Figure 1 materials-16-04363-f001:**
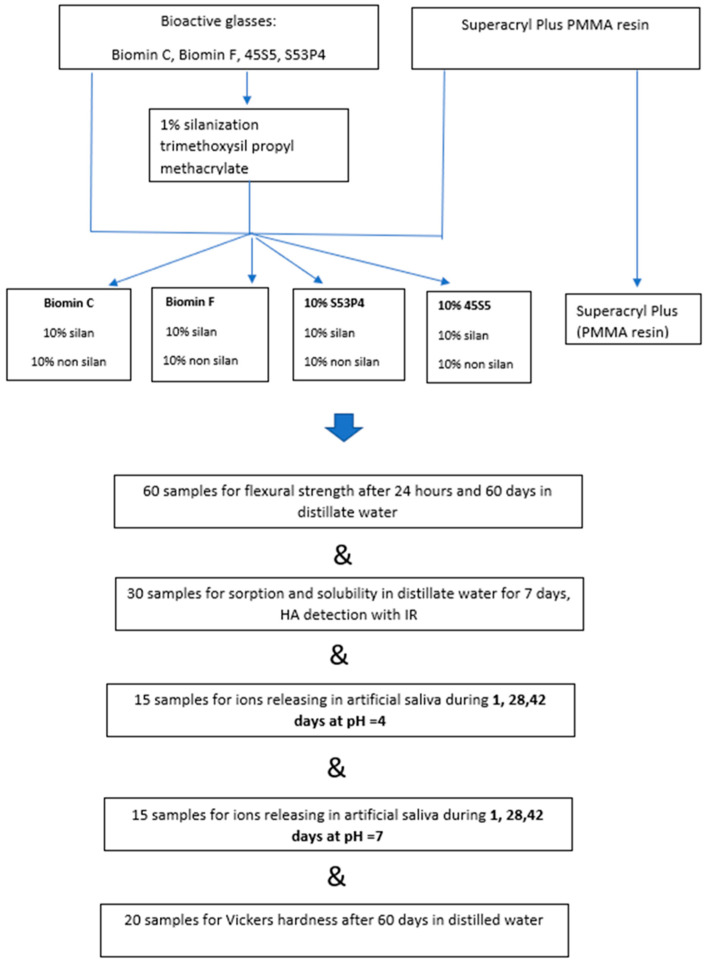
Scheme of sample preparation and testing in this work.

**Figure 2 materials-16-04363-f002:**
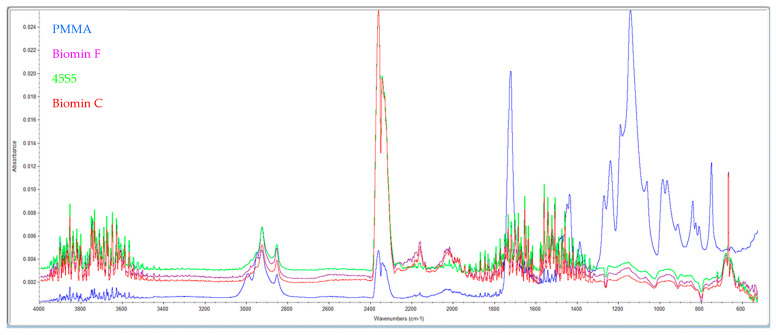
Shows the IR spectra of pure PMMA (blue) and samples with the addition of 53P4 (purple), 45S5 (green), and Biomin F (red) glasses after 7 days in distilled water.

**Table 1 materials-16-04363-t001:** Raw composition of bioactive glasses, according to the bioactive glass manufacture Cera Dynamic.

	SiO_2_	P_2_O_5_	CaO	Na_2_O	CaF_2_	CaCl_2_
S53P4	53.8%	1.7%	21.8%	22.7%	0	0
Biomin F	36.0–40.0%	4–6%	28.0–30.0%	22.0–24.0%	1.5–3.0%	0
45S5	46.1%	2.6%	26.9%	24.4%	0	0
Biomin C	30.3–31.8%	5.0–5.3%	44.1–46.3%	0	0	16.7–20.6%

**Table 2 materials-16-04363-t002:** Flexural strength after storage in distilled water for 24 h and 60 Days Biomin F.

			Biomin F [MPa]	Biomin C [MPa]	53P4 [MPa]	45S5 [MPa]	Resin [MPa]
Flexural strength 24 h			78.13 ± 3.27 *	78.69 ± 5.72 *	80.75 ± 2.41	79.32 ± 2.24	86.5 ± 1.98
Flexural strength 60 days			70.74 ± 1.39 ^®^	69.88 ± 1.73 ^®^	69.88 ± 1.73 ^®^	69.2 ± 2.10 ^®^	79.30 ± 2.55 ^®^

* Statistically significant values at confidence level *p* < 0.05. For samples modified with glass versus resin. ^®^ Statistically significant values at confidence level *p* < 0.01. For samples modified with glass versus resin.

**Table 3 materials-16-04363-t003:** Sorption and solubility of acrylic samples in distilled water for 7 days at 37 °C.

	Biomin F [µg/mm^3^]	Biomin C [µg/mm^3^]	53P4 [µg/mm^3^]	45S5 [µg/mm^3^]	Resin [µg/mm^3^]
Sorption	16.10 ± 2.23	1.49 ± 3.06	19.15 ± 2.37 *	16.85 ± 3.04	14.05 ± 1.08
Solubility	0.85 ± 0.29	2.60 ± 0.88	1.20 ± 0.24	1.00 ± 0.18	1.10 ± 0.34

* Statistically significant at *p* < 0.05 for sample 53P4 vs. pure PMMA.

**Table 4 materials-16-04363-t004:** Ion release by samples of resins modified with bioactive glasses during 1, 28, and 42 days in acidic environment, pH = 4 and neutral pH = 7.

	Ca	P	Si	F
	[mg/L]	[mg/L]	[mg/L]	[mg/L]
**Blank**				
pH 4 0	0	8.218 ± 1.23	0.00	0.00
pH 7 0	0	23.18 ± 3.48	0.00	0.00
**Biomin F**				
pH4 1 day	1.97 ± 0.3	33.12 ± 4.97	15.69 ± 2.35	3.0 ± 0.45
pH4 28 days	0.57 ± 0.09	30.91 ± 4.64	27,26 ± 4,09	3.05 ± 0.46
pH4 42 days	0.62 ± 0.09	30.47 ± 4.35	22.9 ± 3.44	3.1 ± 0.47
pH7 1 day	1.24 ± 0.19	35.40 ± 5.31	11.72 ± 1.76	2.93 ± 0.44
pH7 28 days	0.93 ± 0.14	32.56 ± 4.88	19.26 ± 2.89	3.29 ± 0.49
pH7 42 days	0.74 ± 0.11	31.58 ± 4.74	20.85 ± 3.13	3.09 ± 0.42
**Biomin C**				
pH4 1 day	16.49 ± 2.47	32.57 ± 4.88	18.28 ± 2.74	0.00
pH4 28 days	40.39 ± 6.06	20.38 ± 3.06	31.78 ± 4.77	0.00
pH4 42 days	41.23 ± 6.19	26.43 ± 3.96	33.63 ± 5.04	0.00
pH7 1 day	13.84 ± 2.08	33.17 ± 4.98	8.90 ± 1.33	0.00
pH7 28 days	26.90 ± 4.04	15.26 ± 2.29	25.32 ± 3.80	0.00
pH7 42 days	40.16 ± 6.02	17.50 ± 2.62	33.63 ± 5.04	0.00
**45S53**				
pH4 1 day	2.78 ± 0.42	35.44 ± 5.32	19.33 ± 2.9	0.00
pH4 28 days	3.35 ± 0.50	31.21 ± 4.68	74.12 ± 11.12	0.00
pH4 42 days	1.61 ± 0.24	27.24 ± 4.09	65.54 ± 9.83	0.00
pH7 1 day	1.07 ± 0.16	34.61 ± 0,5.19	17.60 ± 2.64	0.00
pH7 28 days	0.27 ± 0.04	30.49 ± 4.57	43.90 ± 6.59	0.00
pH7 42 days	1.04 ± 0.16	29.30 ± 4.40	53.16 ± 7.97	0.00
**53P4**				
pH4 1 day	2.73 ± 0.41	31.61 ± 4.74	23.20 ± 3.48	0.00
pH4 28 days	1.90 ± 0.29	25.83 ± 3.87	46.52 ± 6.99	0.00
pH4 42 days	1.14 ± 0.17	27.32 ± 4.10	49.23 ± 7.38	0.00
pH7 1 day	0.24 ± 0.03	32.72 ± 4.91	22.45 ± 3.37	0.00
pH7 28 days	0.29 ± 0.04	28.83 ± 4.32	34.61 ± 5.19	0.00
pH7 42 days	0.28 ± 0.04	24.95 ± 3.72	45.30 ± 6.80	0.00
**PMMA**				
pH4 1 day	0.35 ± 0.05	23.94 ± 3.59	0.00	0.00
pH4 28 days	0.38 ± 0.06	26.22 ± 3.93	0.00	0.00
pH4 42 days	0.53 ± 0.08	22.51 ± 3.38	0.00	0.00
pH7 1 day	0.19 ± 0.03	19.80 ± 2.97	0.00	0.00
pH7 28 days	0.25 ± 0.04	30.95 ± 4.64	0.00	0.00
pH7 42 days	0.71 ± 0.11	29.38 ± 3.64	0.00	0.00

**Table 5 materials-16-04363-t005:** Vickers hardness for a sample with 20% glass content stored for 60 days in distilled water.

		Vickers Hardness (HV)	SD
Biomin F	(A)	13.20	0.74
S53P4	(B)	15.36 ^BE^	0.60
45S5	(C)	14.65 ^CE^	0.55
Biomin C	(D)	14.31 ^DE^	0.32
PMMA	(E)	12. 87	0.27

^BE, CE, DE^. Statistically significant values, relative to the pure PMMA (E) sample for the confidence level of confidence *p* < 0.01.

## Data Availability

Not applicable.

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
