# Peer review of "Bioactive Glass-Enhanced Resins: A New Denture Base Material"

_materials, 2023, doi:10.3390/ma16124363_

Round 1

Reviewer 1 Report

Dear Authors

Congratulations on the clinical contribution you have developed with this type of ion releasing materials. You have indeed focused it on the use of prosthetic materials, but even this could be used in the future in endodontic cements, in orthodontics as luting materials, etc. It is a very broad field with many clinical applications. 

It really is a very laborious work in the department but I encourage to carry out these studies of cellular biocompatibility so that its application in the clinic becomes more and more real.

I have only commented on some minor aspects, which appear in the attachment in yellow, but they are minimal.

Sincerely

Author Response

Dear Professor

Thank you very much for your positive reviews. If in the future we could help in some research in the field of bioactive glasses, we are of course open to cooperation.

  All comments have been added to the text

Comment 1

This is not result: These findings suggest that the modified acrylic resins can prevent demineralization and promote hydroxyapatite formation in residual dentition.-

this task has been removed from the abstract, thank you.

Comment 2

This sentence is not objective of the study. By examining these properties, the paper contributes valuable knowledge to the field of dental materials research and expands the current understanding of bioactive glass-containing acrylic resins

This sentence has been removed from the text, thank you.

Comment 3

Please indicate manufacture of each product.

the sentence has been corrected, we added the producer

Table 1. Raw composition of bioactive glasses, according to the bioactive glass manufacture Cera Dynamic.

 Comment 4

 Vancouver style for reference

This task has been improved: These authors have concluded that acrylic resin can release fluoride ions for 56 days for glasses [31] or 15 days for dopped ions to acrylic resins [32-34].

We are grateful to the Reviewer and the Editor for all the remarks and suggestions that helped to improve our manuscript. We have included additional paragraphs to the revised manuscript according to the recommendations. We hope that with these revisions, our manuscript serves better the readers of the Journal than the initially submitted version would, and once more, we thank the Reviewers for their work

Reviewer 2 Report

Dear authors, congratulations for the interesting topic addressed. To facilitate the understanding of some notions, please specify the following:

-does the addition of fluoride mainly consider the antibacterial effect?

- does hydroxylapatite only have the role of improving the surface properties?

- is the obtained material more stable in the oral environment and easier to clean?

Author Response

Dear Professor

thank you for your positive review. We've added all your comments to the text:

note 1

-does the addition of fluoride consider mainly the antibacterial effect?

  In the discussion part, we added the sentence:

Fluoride ions released in such quantities may have a bactericidal effect

Note 2

- does hydroxyapatite only have the role of improving the surface properties?

We have added an explanatory sentence to the discussion section:

  Hydroxyapatite can also form not only on the surface of the modified denture, but also on the residual dentition, which is in direct contact with the prothesis, which prevents demineralization of the enamel

Note 3

- is the obtained material more stable in the oral environment and easier to clean?

This is a very important question. good point. At the moment, we do not know how this material will behave, for example, after a year. We need to investigate this in further tests. This sentence has been moved to the discussion section

We are grateful to the Reviewer and the Editor for all the remarks and suggestions that helped to improve our manuscript. We have included additional paragraphs to the revised manuscript according to the recommendations. We hope that with these revisions, our manuscript serves better the readers of the Journal than the initially submitted version would, and once more, we thank the Reviewers for their work

Reviewer 3 Report

Dear authors,

I carefully read your work and I think that it has to go under major revision.
Here below, I suggest you some criticisms:

- please check the article again and try to refer for scientific and prosthetic terms to the GPT

- the abstract has to follow the author guidelines: The abstract should be a total of about 200 words maximum. The abstract should be a single paragraph and should follow the style of structured abstracts, but without headings

- it is not clear the aim of the study from the abstract. In addition, I suggest to rewrite it in order to be more equilibrate in each section: 1 line of background, 3 of methods and all the other lines are for results and conclusion

- the phrase "These findings suggest that the modified acrylic resins can prevent demineralization and promote hydroxyapatite formation in residual dentition" is a consideration, not a result. It is also repeated in the abstract itself

- The introduction section is well developed with logics but is too long. There are too many examples of approaches that try to modify the surfaces of the material and they are so extended. You can sintetize those sections.

- as you talk about denture base in order to scientific background related to this research topic I would suggest this publication to cite (https://doi.org/10.3390/prosthesis4020015)

- For materials and methods, it is not specified how the sample size was calculated, if there is a control group and how it was fabricated. It seems that the control group is specified later in the text but only in few paragraphs. It would also be important to specify how many elements each control group is composed of.

- Result section is well-done but has to respect the previous comment

- The discussion is well-structured. However, I would also pay attention to the current method of denture base fabrication (milling or 3D printing) and suggest some possible further evaluation of the production method relatively to this new prosthetic material

Author Response

Dear Professor

thank you for your positive review. We've added all your comments to the text:

Note 1

- please check the article again and try to refer for scientific and prosthetic terms to the GPT.

The text has been checked once again for the terminology used

Note 2

-the abstract has to follow the author guidelines: The abstract should be a total of about 200 words maximum. The abstract should be a single paragraph and should follow the style of structured abstracts, but without headings

The abstract has been corrected. It currently counts 188 words. It is one paragraph.

Note 3

-it is not clear the aim of the study from the abstract. In addition, I suggest to rewrite it in order to be more equilibrate in each section: 1 line of background, 3 of methods and all the other lines are for results and conclusion

This part has been corrected. One sentence background, 3 sentences materials and methods and the rest results and conclusions

Note 3

- the phrase "These findings suggest that the modified acrylic resins can prevent demineralization and promote hydroxyapatite formation in residual dentition" is a consideration, not a result. It is also repeated in the abstract itself

Thank you for your comment, this sentence has been removed from the abstract

Note 4

-The introduction section is well developed with logics but is too long. There are too many examples of approaches that try to modify the surfaces of the material and they are so extended. You can sintetize those sections

The Introduction part has been shortened. However, as the authors, we wanted to briefly characterize all the achievements of acrylic surface modification, so that the reader could find more information and also refer to the publications of other authors. In addition, to make it easier for other researchers to get to know the whole issue, and what else can be done for this purpose. Kind of an open forum for ideas and discussion.

Note 5.

- as you talk about denture base in order to scientific background related to this research topic I would suggest this publication to cite (https://doi.org/10.3390/prosthesis4020015)

This important publication explaining the different types of prostheses has been added

Note 6

For materials and methods, it is not specified how the sample size was calculated, if there is a control group and how it was fabricated. It seems that the control group is specified later in the text but only in few paragraphs. It would also be important to specify how many elements each control group is composed of.

Part materials and methods has been modified and explained in more specific way, thank you very much:

Samples of acrylic resin polymerized hot curing method by the addition of bioactive glasses (totally 104 samples) were used for the tests. The same resin for making denture base (totally 30th sample) polymerized according to the manufacturer's recommendations was used as a reference sample. A detailed description of the sample’s preparation is given below and Figure 1.

Note 7

- Result section is well-done but has to respect the previous comment

Part of the results has been re-examined and improved according to your suggestions

Note 8

-The discussion is well-structured. However, I would also pay attention to the current method of denture base fabrication (milling or 3D printing) and suggest some possible further evaluation of the production method relatively to this new prosthetic material

Part of the results has been re-examined and improved according to your suggestions

The use of this type of bioactive glass in further research can be extended to materials used in CAD CAM technology and 3D printing.

We are grateful to the Reviewer and the Editor for all the remarks and suggestions that helped to improve our manuscript. We have included additional paragraphs to the revised manuscript according to the recommendations. We hope that with these revisions, our manuscript serves better the readers of the Journal than the initially submitted version would, and once more, we thank the Reviewers for their work

Round 2

Reviewer 3 Report

I have no more comments for the authors